# MFRGN: Multi-scale Feature Representation Generalization Network For Ground-to-Aerial Geo-localization

## ABSTRACT

Cross-area evaluation poses a significant challenge for ground-to-aerial geo-localization (G2AGL), in which the training and testing data are captured from entirely distinct areas. However, current methods struggle in cross-area evaluation due to their emphasis solely on learning global information from single-scale features. Some efforts alleviate this problem but rely on complex and specific technologies like pre-processing and hard sample mining. To this end, we propose a pure end-to-end solution, free from task-specific techniques, termed the Multi-scale Feature Representation Generalization Network (MFRGN) to improve generalization. Specifically, we introduce multi-scale features and explicitly utilize them for G2GAL. Furthermore, we devise an efficient global-local information module with two flows to bolster feature representations. In the global flow, we present a lightweight Self and Cross Attention Module (SCAM) to efficiently learn global embeddings. In the local flow, we develop a Global-Prompt Attention Block (GPAB) to capture discriminative features under the global embeddings as prompts. As a result, our approach generates robust descriptors representing multi-scale global and local information, thereby enhancing the model's invariance to scene variations. Extensive experiments on benchmarks show our MFRGN achieves competitive performance in same-area evaluation and improves cross-area generalization by a significant margin compared to SOTA methods.

## CCS CONCEPTS

• **Computing methodologies** → **Visual content-based indexing and retrieval**; **Matching**; **Image representations**.

## KEYWORDS

Cross-view image geo-localization, image matching, image retrieval, information representation

## 1 INTRODUCTION

Ground-to-Aerial Geo-Localization (G2AGL) aims to determine the real-world geographic location of ground/street-view images (as queries) by matching with GPS-tagged aerial/satellite-view images (as references) covering the same geographic region. It has a great potential for accurate location with noisy GPS [4, 46], autonomous driving [16, 32], automatic navigation [17, 44], augmented reality [27] and object localization [42]. However, the drastic differences

*ACM MM, 2024, Melbourne, Australia*
© 2024 Copyright held by the owner/author(s). Publication rights licensed to ACM.
ACM ISBN 978-x-xxxx-xxxx-x/YY/MM
https://doi.org/10.1145/nnnnnnn.nnnnnnn

**Unpublished working draft. Not for distribution.**

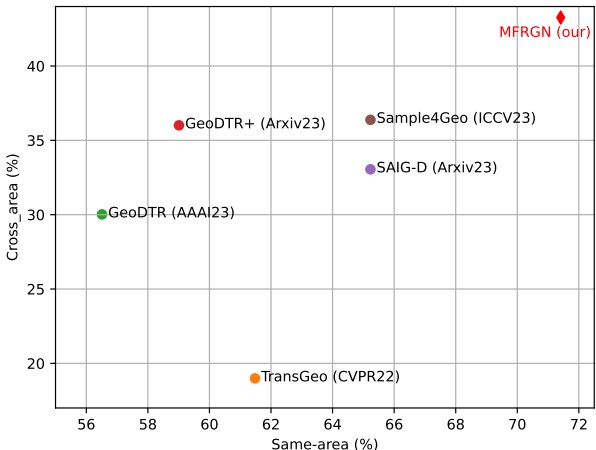

Figure 1: Comparison of R@1 accuracy between recent methods without using task-specific techniques on VIGOR benchmark, encompassing both polar transformation [31, 45] and hard negative mining [7]. Our proposed MFRGN achieves the SOTA same-area and cross-area evaluation for G2AGL.

in viewpoint and appearance make the two-view images distinct features, and even some contents of the one-view images are invisible to the other. Consequently, G2AGL is an extremely challenging problem.

Existing approaches [5, 12, 13, 20, 43] typically learn a common representation across different views to convert image geo-localization into an image-to-image matching/retrieval task. However, these methods achieve high same-area performance, in which the training and testing data are captured from identical areas, but low cross-area evaluation, where data are collected from completely distinct areas. For example, TransGeo [52] demonstrates excellent same-area retrieval accuracy, e.g. ~85% top-1 recall (R@1) accuracy on CVACT [21] benchmark, but notably low R@1 across different one, e.g. only 17.45% when training on CVACT (mostly included urban scenery in Canberra, Australia) and testing on CVUSA [43, 47] (collected from across the United States). Others attempt to improve cross-area generalization by complex pre-processing techniques[45, 49], e.g., the polar transformation relying on a center alignment assumption between two views, which often unmeting in real-life settings. Additionally, elaborate hard sample mining techniques like dynamic similarity sampling strategy [7] and contrastive hard sample generation [48] have been explored. To make G2AGL more practical, some methods try to get rid of these task-specific technologies, but there remains a notable performance gap between same- and cross-area evaluation for these algorithms without these techniques on the challenging VIGOR [53] benchmark, as shown in Fig. 1.

Motivated by these observations, this paper makes concerted efforts to improve cross-area performance by devising an efficiently pure structure devoid of specific techniques. As illustrated in Fig. 2, the basic structures of the existing methods can be broadly divided into two categories: those based on CNN or Vision Transformer (ViT) [38], and those based on a combination of CNN and ViT. However, single-scale features considered by most methods may be insufficient for new or changed scenes, which limits the final performance. It's noteworthy that features conducive to localization may be distributed across different scales due to variations in distance and resolution of image acquisition. On the other hand, since images from nearby yet distinct geographic locations often share common properties like street layout or building type, existing methods focusing only on learning global context fail to distinguish them, leading to matching with the same reference. This can be mitigated by incorporating local information, e.g. the diverse colors and textures found in two buildings. As a matter of fact, humans typically employ a "rough screening through global context, then fine matching through local details" mechanism when pairing two images. Hence, we introduce multi-scale features and jointly learn global and local information from them to bolster robustness in feature representations.

In this paper, we introduce a Multi-scale Feature Representation Generalization Network (MFRGN) aimed at enhancing generalization for real-world applications. A simplified structure diagram of MFRGN is depicted in Fig. 2 (c), which differs from the two existing architectures (a) and (b). Firstly, we leverage a pre-trained CNN backbone to acquire multiple feature maps with richer information across different scales. Subsequently, we devise an efficient global-local information representation module to concurrently learn global and local representation. In this module, we introduce ViT-based Self and Cross Attention Module (SCAM) to efficiently learn multi-scale global context representations, and propose a CNN-based Global-Prompt Attention Block (GPAB) to encourage the model to learn distinctive local features under the global representations as prompts. Furthermore, to alleviate model burden and feature redundancy on multiple scales, we introduce a lightweight Transformer encoder and a concise Pyramid Pooling Sampling (PPS) strategy. As a result, MFRGN improves the abilities in terms of feature representation and model generalization by simultaneously learning global and local embeddings across multi-scale features. Our code will be released.

The main contributions of this work are as follows:

1) We explicitly utilize multi-scale features and propose MFRGN to enhance the model's representational capability, thereby improving cross-area generalization.

2) We simultaneously consider global and local information (possibly for the first time in G2AGL's task) by a well-designed dual-flow structure based on ViT and CNN.

3) We present lightweight SCAM to efficiently model global dependencies and propose GPAB to boost local representation with learned global embedding as prompts, both of which are in the multi-scale case.

4) Our framework is simple and efficient, and purely end-to-end requiring no neither complex pre-processing steps relying on strict assumptions, nor specific sample mining technologies. Our method

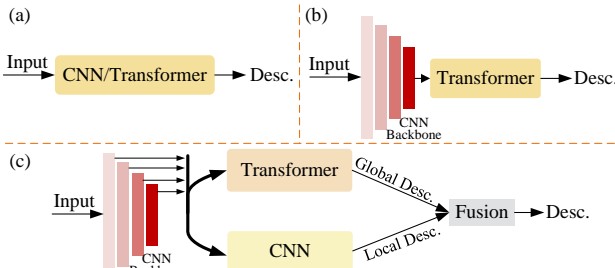

Figure 2: Basic structures for G2AGL. Input is ground-view images or aerial-view images, and Desc. is descriptor. Exitsing structures, (a) and (b), aim to learn global descriptors. The proposed novel structure (c) extracts multi-scale features and explicitly utilizes them to learn global and local representation for better cross-area performance in G2AGL.

outperforms previous works in generalization evaluation by a large margin.

## 2 RELATED WORK

### 2.1 CNN & Transformer structures

Early methods [2, 39] based on handcrafted features [3, 24] face challenges due to the large viewpoint and appearance differences between queries and references, resulting in extremely low retrieval accuracy. Subsequent efforts [7, 11–13, 17] shift towards deep learning-based methods to extract powerful deep features. Workman et al. [43] demonstrate the superiority of deep features over hand-crafted ones by utilizing a deep CNN network, achieving high accuracy. Building upon this pioneering work, Vo el et. [40] design Siamese-like CNN for learning image features, while Zhai et al. [47] employ a VGG16 architecture to predict a dense pixel-level segmentation of the ground image. Further advancements include CRN [14], which learns contextual feature representations by reweighting CNN to produce a spatial weighting mask. CVM-Net [11] utilizes a full CNN to extract local features and a NetVLAD [1] layer to get the global descriptor. GeoCapNet [35] learns the spatial feature hierarchies to enhance representation based on ResNet [10] and capsule network [30]. Recently, Vision Transformer (ViT) [38] has achieved marvelous success in drawing global dependencies. Due to the significant computational burden of directly applying ViT, L2LTR [45] combines CNN and ViT (i.e. hybrid ViT) to reduce visual ambiguity by extracting global contextual descriptors, where CNN learns high-level semantic features from input images and ViT encodes them to produce global representations. Inspired by [31], GeoDTR [49] introduces a geometric layout extractor based on ViT to generate attention maps from high-level semantics features of the CNN backbone. TransGeo [52] proposes a pure ViT-based model to learn global context directly from input images by using attention-guided non-uniform cropping, and achieves outstanding results in same-area evaluation, yet poor results in cross-area evaluation.

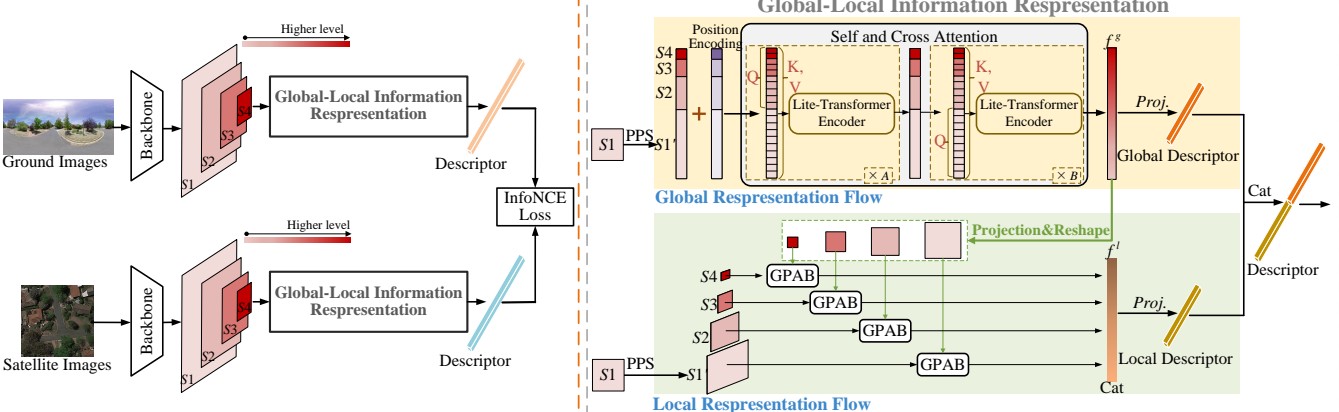

**Figure 3: Pipeline of MFRGN, a Siamese network with two identical branches. In each branch, firstly, images (ground/satellite-view) are fed into a CNN backbone to extract multi-scale features. Then, we input them into a dual-flow structure, namely global-local information representation. The global representation flow learns global representations ($f^g$) using a lightweight Self and Cross Attention Module (SCAM), while the local representation flow learns local representations ($f^l$) by four parallel Global-Prompt Attention Block (GPAB) with $f^g$ as prompt. Finally, $f^g$ and $f^l$ are individually processed by a linear projection ($Proj.$) and then concatenated them to output the final descriptor.**

## 2.2 Task-specific technologies

To bridge the large domain gap between panoramic and satellite images, some methods specialize in designing specific techniques for G2AGL. For example, [29] introduces conditional GAN (cGAN) to synthesize overhead-view images from ground-view images, while [36] does the opposite. However, their achievements of high accuracy depend on the generated successful results of the cGAN. Alternatively, other methods [31, 33, 36] have adopted polar transformation to tackle the cross-view domain gap by leveraging the center alignment assumption: a reference image exactly centers at the location of any query image. For example, Shi et al. [31] apply this transformation to preprocess aerial images into ground-view images, which are similar to the domain of the street images, resulting in remarkably high retrieval accuracy. DSM [33] directly estimates the orientation alignment of the two-view images under this assumption. Later, numerous methods [25, 34, 45, 48] employ this transformation as a pre-processing technique. Moreover, some methods tend to mine samples. GeoDTR+ [48], a recent outstanding method aimed at improving cross-area evaluation, introduces a hard sample mining strategy to generate additional challenging samples in training batches. Sample4geo [7] develops GPS-based sampling and dynamic similarity sampling strategies to sample hard negatives, outperforming previous SOTA results without polar transformation. Yet, it still relies on an assumption that both query and reference images have GPS tags in the raw data, which are unmet in the real world. In contrast, some works have proposed datasets that do not adhere to these limitations. For instance, the VIGOR [53] dataset allows ground images' GPS coordinates to be at any location of aerial images, while the University-1652 [51] dataset is primarily proposed for drone-to-aerial geo-localization but also provides ground images collected from ground cameras without GPS tags.

## 3 METHODOLOGY

In this section, we present the proposed Multi-scale Feature Representation Generalization Network (MFRGN), aiming to enhance generalization for ground-to-aerial geo-localization. MFRGN is a Siamese neural network with two identical branches. Each branch consists of two main steps: multi-scale feature extraction (Sec. 3.1) and feature representation with a global representation flow (GRF) to learn global embeddings (Sec. 3.2), along with a local representation flow (LRF) to reinforce local feature representation with the global embeddings as prompts (Sec. 3.3). Consequently, we obtain a global descriptor from the GRF and a local descriptor from the LRF, then concatenate them together as the final output descriptor for similarity calculation. An overview of our MFRGN is presented in Fig. 3.

Following [7, 9, 19], we utilize symmetric InfoNCE loss based on Noise-Contrastive Estimation (NCE) [26, 28] to train our model.

$$\mathcal{L}(q, R)_{\text{InfoNCE}} = -\log \frac{\exp(q \cdot r_+/\tau)}{\sum_R^{i=0} \exp(q \cdot r_i/\tau)} \quad (1)$$

where $q$ denotes an encoded query, and $R$ is a set of encoded references. The InfoNCE loss calculates cosine similarity and is low when $q$ and the positive match $r_+$ are similar, and high when they are dissimilar. The temperature parameter $\tau$ is learnable.

## 3.1 Multi-scale Feature Extraction

Due to the distance at which the images were taken and the different resolutions of the images captured, which are more pronounced across different areas, features favorable for localization are distributed across different scales. Therefore, we extract features at multiple scales as opposed to a single scale in most existing methods [31, 45, 49]. We employ a pre-trained CNN (e.g. RetNet [10], ConvNeXt [22]) as our backbone with multiple stages for multi-scale feature extraction.

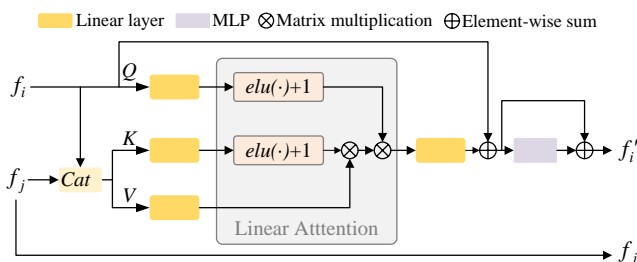

**Figure 4: Lite-Transformer encoder (LTE).** $f_i$ is $F_H$ (= S2 ∼ S4) when $f_j$ (= S1) is $F_L$ and vice versa. *Cat* represents a concatenation operation. $elu(\cdot)$ denotes the exponential linear unit activation function.

Given that the output feature maps of the initial backbone stages (e.g. , stage0, stage1) tend to be large, leading to a substantial number of tokens if a ViT-like architecture is subsequently involved, we opt the output maps from the final three backbone stages, denoted as S1, S2, S3, as shown in Fig. 3. S4 is acquired by further downsampling S3 by a ratio of 0.5. Ranging from S1 to S4, the feature scale progressively increases, capturing richer semantic information conducive to global matching. Conversely, lower-level features contain more detailed information beneficial for localized fine matching.

## 3.2 Global Representation Flow

Most approaches [21, 47–49, 52] for G2AGL achieve better results by generating global descriptors, which indicates the significance of global information. Leveraging the notable capability of ViT in capturing global dependencies, we adopt ViT to construct global representation. We designate S1 ∼ S4 as the Q (query), K (key), and V (value) for each ViT encoder. In this configuration, cross-attention can be regarded as retrieving corresponding features among different scale features when the tokens of Q, K, and V come from different scale features. On the other hand, self-attention resembles retrieving corresponding features within the same scale features when tokens stem from the same scale features. Therefore, we propose a ViT-based Self and Cross Attention Module (SCAM) (shown in Fig. 3), which effectively exploits both self-attention and cross-attention mechanisms to capture global dependencies across different scale features, contributing to more effective feature representation.

However, it is impractical to directly utilize features S1 ∼ S4 for ViT. This is because these multi-scale features would generate a large number of tokens, leading to excessively high model complexity, even rendering training infeasible. Inspired by Lite-DETR [18], we present features S2 ∼ S4 as high-level features ($F_H$) and S1 as low-level features ($F_L$). The number of tokens in $F_H$ accounts for 6% ∼ 33% of $F_L$, significantly reducing computational costs. Due to global descriptors primarily focusing on global semantics, we update $F_H$ at a higher frequency. As shown in Fig. 3 and 4, we firstly utilize $F_H$ as the query $Q$ and concatenate $F_L$ with $F_H$ as the key and value ($K = V$), which are then inputted into Lite-Transformer Encoder (LTE) for updating $F_H$. The update process of $F_H$ is as

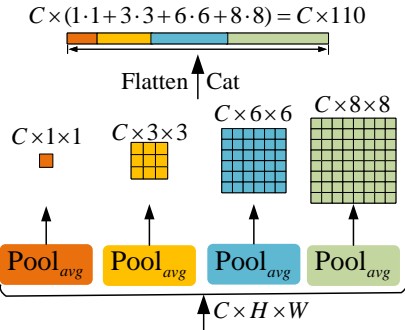

**Figure 5: Pyramid pooling sampling (PPS).**

follows:

$$Q = F_H, K = V = Concat(F_H, F_L)$$

$$\begin{cases} Output1 = F'_H = Q' = LTE(Q, K, V) \\ Output2 = F_L \end{cases} \quad (2)$$

where *Concat* represents concatenating all scale features, $F_H$ and $F_L$ are the initial input features, and $F'_H$ are the updated features. $F_H$ is updated $A$ times.

$F_L$ contains only a small amount of global semantic information; therefore, it is updated only a few times. The updating process of $F_L$ is as follows:

$$Q = F_L, K = V = Concat(F_L, F'_H)$$

$$\begin{cases} Output1 = F'_L = LTE(Q, K, V) \\ Output2 = F'_H \end{cases} \quad (3)$$

where $F_L$ represents the original low-level features, $F'_H$ represents the features updated from the previous iteration of $F_H$, and $F'_L$ represents the updated features. $F_L$ is updated for $B$ iterations. The output of SCAM is represented as $f^g$(= $Concat(FL', FH')$), followed by linear projection to generate the global descriptor (Desc$^g$).

However, despite the above steps taken to alleviate model costs, the computational demands of the ViT model remain high under multi-scale settings. To address this issue, we employ two strategies.

**Strategy 1**: Considering that the dot-product attention with $O(N^2)$ complexity in the vanilla ViT, we replace it with the linear attention [15] with only $O(N)$ complexity, referred to as the Lite-Transformer encoder (Fig. 4). Formally, the linear attention substitutes the vanilla dot-product with an alternative attention function $\phi(Q)(\phi(K)^T V)$, where $\phi(\cdot) = elu(\cdot)+1$, and $elu(\cdot)$ denotes the exponential linear unit [6]. Fig. 4 illustrates the computation graph of this operation.

**Strategy 2**: Given that the lower-level features occupy the majority of tokens and contain many redundant features, we employ a pyramid pooling sampling (PPS) strategy [50] to sample $n$ ($n \ll H \times W$) representative feature points from S1. Fig. 5 shows the default configuration: four average pooling operations are initially performed to obtain four feature maps with sizes of $1 \times 1$, $3 \times 3$, $6 \times 6$, and $8 \times 8$, respectively. Subsequently, we flatten the four maps and then concatenate them together to obtain a sampled feature

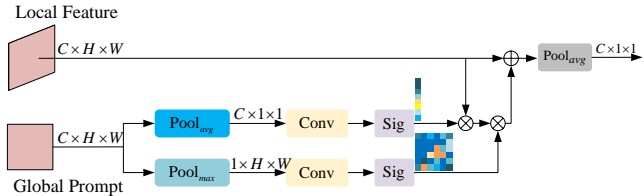

**Figure 6: The global-prompt attention block (GPAB).**

map with a length of 110. For G2AGL, we set $n$ to approximately twice the total number of tokens of $S2 \sim S4$.

## 3.3 Local Representation Flow

Based on the observation: a human typically confirms whether two images belong to the same scene by their overall appearance, and then verifies their correspondence-specific details. Generally, the "overall" refers to global semantics, while the "detail" refers to local textures. There is an interesting fact that local results are influenced by global confirmation, meaning that in this case where rough global matching is known, the local results would be more adept at capturing discriminative features conducive to localization. Therefore, we consider global information into the LRF. Specifically, we use the global representation ($f^g$) from the GRF as prompts to guide the learning of local representation (denoted as $f^l$), shown in Fig. 3. Due to the inconsistency between $f^g$ and $f^l$, directly fusing them may hinder information expression. We project $f^g$ into the local space, then reshape it to match the sizes of $S1 \sim S4$, resulting in four global prompts ($f_1^g \sim f_4^g$). Subsequently, we propose a Global-Prompt Attention Block (GPAB) to achieve prompt learning, as shown in Fig. 6. Each global prompt $f_i^g$ undergo operations according to Eq. 4, resulting in channel-wise and spatial-wise weights $F_i^{ch}$ and $F_i^{ch}$, respectively.

$$\begin{cases} F_i^{ch} = \text{Sig}(\text{Conv}(\text{pool}_{avg}(f_i^g))) \\ F_i^{sp} = \text{Sig}(\text{Conv}(\text{pool}_{max}(f_i^g))) \end{cases}, \ i = 1, 2, 3, 4 \quad (4)$$

where $\text{pool}_{avg}$ represents average pooling along the channel direction, $\text{pool}_{max}$ represents max pooling along the spatial direction, Conv denotes the 2D convolution operation, and Sig is the Sigmoid activation function.

We utilize $F_i^{ch}$ and $F_i^{sp}$ to guide $S_i$ to learn local discriminative information, ultimately leading to local representation $f_i^l$:

$$f_i^l = \text{Pool}_{avg}(S_i \cdot F_i^{ch} \cdot F_i^{sp} + S_i), \ i = 1, 2, 3, 4 \quad (5)$$

All $f_i^l$ are concatenated to $f^l$ that then generates local descriptor $\text{Desc}^l$ by a linear projection. Similar to sampling strategies described in Sec. 3.2, we also use the PPS to sample $S1$ in LRF.

## 4 EXPERIMENTS

### 4.1 Experiments settings

**Datasets.** Both CVUSA [43, 47] and CVACT [21] are cross-view datasets for ground-to-aerial geo-localization, each containing 35,532 pairs for training and 8,884 pairs for testing. CVUSA collects images from across the United States, while CVACT mainly covers urban scenery in Canberra, Australia. The two datasets are used for

one-to-one matching, i.e. every satellite-view image has one corresponding street-view image, and share an assumption of center alignment: all pairs are aligned with similar spatial localization and orientation. For the two datasets, we use $256 \times 256$ for the aerial view and $128 \times 256$ for the ground view as input image size. VIGOR [53] is a challenging and large-scale dataset for one-to-more mapping (every reference covers 2~4 queries), which is a more real-life setting. VIGOR collects 105,124 street panoramas and 90,618 satellite images from four cities, New York, Chicago, Seattle, and San Francisco. VIGOR provides two evaluation settings, i.e. same-area (training and testing on all cities) and cross-area (training on New York and Seattle, testing on San Francisco and Chicago) evaluation for comprehensive performance in G2AGL. Exceptionally, the geographical location of a panorama can correspond to any location in the aerial image. Also, each satellite-view image has three semi-positive images that cover regions of the street-view image. For VIGOR, we use $320 \times 320$ for the aerial view and $320 \times 640$ for the ground view as input image size.

**Metrics.** Similar to existing methods [7, 8, 45, 49], we choose to use the top-$k$ recall accuracy (denoted as R@$k$) with $k \in \{1, 5, 10, 1\%\}$ for evaluation purposes. R@$k$ assesses the probability of the ground truth reference image within the top $k$ ranked results given a query image. Besides, we also use hit rate to evaluate the retrieval performance of VIGOR following [7]. The hit rate is understood as the R@1 without semi-positive images.

**Implementation Details.** Following [7, 19, 48], we use weight-shared ConvNeXt [22] as our CNN backbone for both views. We set $A = 2, B = 1$ in the SCAM. And, we set the latent dimension to 128 and the feedforward layer dimension to 2048 for each Lite-Transformer encoder with 4 heads. In PPS, we obtain four feature maps with sizes of $1 \times 1, 6 \times 6, 12 \times 12, 21 \times 21$ for satellite view, and $1 \times 1, 3 \times 12, 6 \times 24, 7 \times 36$ for street view, respectively. In the training phase, the model is trained on Nvidia Titan V GPUs for 50 epochs with AdamW [23] optimizer. The label smoothing of InfoNCE loss is set to 0.1, and the batch size to 64. We use a cosine annealing learning rate decay strategy with an initial learning rate of $10^{-4}$. The dimension of the final descriptor is 2048.

### 4.2 Comparison with State-of-the-Art Methods

**Same-area evaluation results.** Tab. 1 presents same-area results of various methods. In the "w/ **Pre-processing**" column, all methods use a pre-cropping technology (i.e. pre-crop the edges of street images in CVACT) to accelerate model fitting, and some of them also use the polar transformation (marked with the †) to pre-process satellite images into ground-view images, which both rely on the center alignment assumption. We observe that MFRGN with only using pro-cropping achieves the best improvement across all four metrics. In the "w/ **Sampling**" column, our method brings a slight performance fluctuation using hard negative sampling (HNS) (requiring the GPS-tagged query images) in [7]. When not using all of the above, our retrieval results surpass all others. Under this condition, an interesting observation is that the performance of our method far exceeds all methods with "**Pre-processing**" and reaches the same level as Sample4Geo with "**Sampling**" on CVUSA, even scores higher in terms of R@1, R@10 and R@1% on CVACT. Besides, MFGAN+ obtains the maximum gain with pro-cropping

**Table 1: Same-area comparison results on CVUSA and CVACT benchmarks. † denotes using polar transformation. The best and second best results are bolded and underlined.**

| Method | CVUSA | | | | CVACT | | | |
|---|---|---|---|---|---|---|---|---|
| | R@1 | R@5 | R@10 | R@1% | R@1 | R@5 | R@10 | R@1% |
| *w/ Pre-processing* | | | | | | | | |
| SAFA [31] | 81.15 | 94.23 | 96.85 | 99.49 | 78.28 | 91.60 | 93.79 | 98.15 |
| SAFA† [31] | 89.84 | 96.93 | 98.14 | 99.64 | 81.03 | 92.80 | 94.84 | 98.17 |
| SAFA†+LPN [41] | 92.83 | 98.00 | 8.85 | 99.78 | 83.66 | 94.14 | 95.92 | 98.41 |
| GeoDTR [49] | 93.76 | 98.47 | 99.22 | 99.85 | 85.43 | 94.81 | 96.11 | 98.26 |
| GeoDTR† [49] | 95.43 | 98.86 | 99.34 | 99.86 | 86.21 | 95.44 | 96.72 | 98.77 |
| GeoDTR+ [48] | 95.05 | 98.42 | 98.92 | 99.77 | 87.76 | 95.50 | 96.50 | 98.32 |
| GeoDTR+† [48] | 95.40 | 98.44 | 99.05 | 99.75 | 87.61 | 95.48 | 96.52 | 98.34 |
| SAIG-D [54] | 96.08 | 98.72 | 99.22 | 99.86 | 89.21 | 96.07 | 97.04 | 98.74 |
| SAIG-D† [54] | 96.34 | 99.10 | 99.50 | 99.86 | 89.06 | 96.11 | 97.08 | 98.89 |
| MFRGN (ours) | **98.24** | **99.56** | **99.72** | **99.88** | **89.51** | **96.96** | **97.75** | **99.01** |
| *w/ Sampling* | | | | | | | | |
| Sample4Geo [7] | 98.68 | **99.68** | **99.78** | **99.87** | **90.81** | **96.74** | **97.48** | **98.77** |
| MFRGN (ours) | **98.69** | 99.58 | 99.67 | 99.82 | 90.54 | 96.12 | 96.84 | 98.37 |
| *w/o Pre-processing & Sampling* | | | | | | | | |
| L2LTR [45] | 91.99 | 97.68 | 98.65 | 99.75 | 83.14 | 93.84 | 95.51 | 98.40 |
| TransGeo [52] | 94.08 | 98.36 | 99.04 | 99.77 | 84.95 | 94.14 | 95.78 | 98.37 |
| Sample4Geo [7] | 97.84 | **99.58** | **99.75** | **99.88** | 87.73 | 96.58 | **97.59** | **98.99** |
| MFRGN (ours) | **98.24** | 99.56 | 99.72 | **99.88** | **88.78** | **96.61** | 97.51 | **98.99** |
| MFRGN+ (ours) | 98.67 | 99.57 | 99.71 | 99.85 | 91.09 | 96.34 | 97.14 | 98.44 |

**Table 2: Cross-area comparison results on CVUSA and CVACT benchmarks. † denotes using polar transformation. The best and second best results are bolded and underlined.**

| Method | CVUSA→CVACT | | | | CVACT→CVUSA | | | |
|---|---|---|---|---|---|---|---|---|
| | R@1 | R@5 | R@10 | R@1% | R@1 | R@5 | R@10 | R@1% |
| *w/ Pre-processing* | | | | | | | | |
| SAFA† [31] | 30.40 | 52.93 | 62.29 | 85.82 | 21.45 | 36.55 | 43.79 | 69.83 |
| GeoDTR [49] | 43.72 | 66.99 | 74.61 | 91.83 | 29.85 | 49.25 | 57.11 | 2.47 |
| GeoDTR† [49] | 53.16 | 75.62 | 81.90 | 93.80 | 44.07 | 64.66 | 72.08 | 90.09 |
| GeoDTR+ [48] | 60.16 | 79.97 | 84.67 | 94.48 | 52.56 | 73.08 | 79.82 | 94.80 |
| GeoDTR+† [48] | 61.17 | 80.22 | 85.45 | 94.56 | 53.89 | 74.56 | 81.10 | 94.93 |
| MFRGN (ours) | **71.56** | **88.98** | **92.06** | **97.12** | **55.32** | **76.64** | **83.14** | **96.15** |
| *w/ Sampling* | | | | | | | | |
| Sample4Geo [7] | 56.62 | 77.79 | **87.02** | **94.69** | 44.95 | 64.36 | 72.10 | 90.65 |
| MFRGN (ours) | **66.06** | **82.79** | 86.58 | 94.47 | **63.34** | **80.54** | **86.22** | **96.81** |
| *w/o Pre-processing & Sampling* | | | | | | | | |
| TransGeo [52] | 37.81 | 61.57 | 69.86 | 89.14 | 17.45 | 32.49 | 40.48 | 69.14 |
| Sample4Geo [7] | 34.88 | 59.68 | 69.28 | 90.02 | 15.78 | 31.13 | 39.40 | 70.04 |
| MFRGN (ours) | **51.61** | **73.91** | **80.55** | **94.24** | **49.18** | **70.09** | **77.42** | **95.04** |
| MFRGN+ (ours) | **79.12** | **91.09** | **93.17** | **96.79** | **69.28** | **84.91** | **89.60** | **97.69** |

and HNS, noting that without polar transformation. We further evaluate various methods without **Pre-processing** and **Sampling** on the highly challenging VIGOR (Tab. 3). Our MFRGN achieves the best scores compared with all methods. These results indicate that MFGRN can learn powerful feature representation that eliminates the need for these task-specific techniques to bridge the domain gap between the two distinct viewpoints, even in challenging scenarios.

**Table 3: Same-area and cross-area comparison results on VIGOR benchmark. All methods do not use "*Pre-processing*" and "*Sampling*".**

| Method | R@1 | R@5 | R@10 | R@1% | Hit rate |
|---|---|---|---|---|---|
| *Cross-area* | | | | | |
| TransGeo [52] | 61.48 | 87.54 | 91.88 | 99.56 | 73.09 |
| GeoDTR [49] | 56.51 | 80.37 | 86.21 | 99.25 | 61.76 |
| GeoDTR+ [48] | 59.01 | 81.77 | 87.10 | 99.07 | 67.41 |
| SAIG-D [54] | 65.23 | - | 88.08 | 99.68 | 74.11 |
| Sample4Geo [7] | 65.23 | 91.62 | **95.85** | **99.85** | 78.77 |
| MFRGN (ours) | **71.41** | **92.02** | 95.07 | 99.82 | **80.59** |
| *Cross-area* | | | | | |
| TransGeo [52] | 18.99 | 38.24 | 46.91 | 88.94 | 21.21 |
| GeoDTR [49] | 30.02 | 52.67 | 61.45 | 94.40 | 30.19 |
| GeoDTR+ [48] | 36.01 | 59.06 | 67.22 | 94.95 | 39.40 |
| SAIG-D [54] | 33.05 | - | 55.94 | 94.64 | 36.71 |
| Sample4Geo [7] | 36.38 | 63.96 | 72.43 | 97.18 | 43.66 |
| MFRGN (ours) | **43.18** | **67.52** | **75.34** | **97.47** | **47.54** |

**Cross-area evaluation results.** We present the cross-area results of various methods to evaluate their generalization capabilities. The CVUSA→CVACT represents training on CVUSA and testing on CVACT, while CVACT→CVUSA represents the opposite. As shown in Tab. 2, our model performs the best results almost in all cases by a large margin. Specifically, our method achieves the SOTA performance by improving from the previous SOTA 37.81% to 51.61% on the CVUSA → CVACT, and 17.45% to 49.18% on the CVACT→CVUSA when no any special techniques are used. For the challenging VIGOR, MFRGN obtains significant cross-area improvements and improves 6.80% R@1 and 3.88% hit rate compared to Samole4Geo (Tab. 3). These considerable improvements can be attributed to the fact that our approach improves representation abilities by jointly learning global and local information from multi-scale features, thereby generalizing to diverse cross-area scenarios.

## 4.3 Ablation Studies

**Impact of components.** Tab. 4 performs a component analysis of MFRGN, such as the GRF ($G$), the LRF with global embeddings as prompts ($L_p$) or with the initial features ($S1 \sim S4$) as prompts ($L_{ini}$), and so on. We use the highest level feature map from the last stage of the CNN backbone (i.e. $S4$) to directly output the descriptor, denoted as *Baseline*. When adding multi-scale features (*MBaseline*) into the *Baseline*, the performance gains are slightly smaller than *Baseline* in the same area but are obviously larger in cross area. We conjecture that multi-scale features could introduce more local details but hamper information conveyance; whereas, in the context of the cross area, these features containing richer and more match-relevant information across different scales help adapt to diverse scenes. Furthermore, our model improves greatly cross-area performance after adding GRF or LRF on *MBaseline*, especially in the former. This proves that global information is more favorable for image-to-image matching than local information. However, jointly learning $G$ and $L_{ini}$ / $L_p$ can further significantly

**Table 4: Component-wise analysis when training on CVACT.**

| Method | R@1 | R@5 | R@10 | R@1% |
|---|---|---|---|---|
| ***CVACT*** | | | | |
| *Baseline* | 83.52 | 95.27 | 96.75 | 98.96 |
| *MBaseline* | 80.76 | 95.36 | 96.84 | 99.01 |
| *MBaseline+$L_{ini}$* | 81.59 | 95.03 | 96.36 | 98.90 |
| *MBaseline+G* | 86.58 | 96.15 | 97.20 | 99.00 |
| *MBaseline+G+$L_{ini}$* | 88.21 | 96.37 | 97.40 | 99.03 |
| *MBaseline+G+$L_p$* | 88.42 | 96.49 | 97.49 | 99.04 |
| ***CVACT→CVUSA*** | | | | |
| *Baseline* | 10.78 | 23.37 | 30.68 | 62.89 |
| *MBaseline* | 16.81 | 33.21 | 43.31 | 72.37 |
| *MBaseline+$L_{ini}$* | 18.07 | 35.21 | 44.68 | 78.13 |
| *MBaseline+G* | 36.61 | 57.97 | 66.51 | 91.31 |
| *MBaseline+G+$L_{ini}$* | 45.54 | 68.35 | 76.15 | 94.90 |
| *MBaseline+G+$L_p$* | 47.13 | 69.07 | 77.51 | 95.48 |

**Table 5: Impact of two lightweight strategies, linear attention (LA), and pyramid pooling sampling (PPS), when training on CVACT. ↑ denotes the performance gain compared to MFRGN without LA and PPS.**

| LA | PPS | FLOPs | Memory | Same-area | Cross-area |
|---|---|---|---|---|---|
| × | × | 22.31G | 1166M | 87.47 | 42.14 |
| ✓ | | 22.31G | 1065M | 88.42 (↑0.95) | 47.13 (↑4.99) |
| ✓ | ✓ | 21.80G | 1050M | 88.78 (↑1.31) | 49.18 (↑7.04) |

improve performance. In this case, $L_p$, namely learning global and local information by an interaction, can obtain better local feature representation, compared to $L_{ini}$, namely LRF not interacting with GRF. Therefore, MFRGN can better bridge the gap not only between two view images but also between two area distributions by learning the rich and robust representation, thus improving the performance and generalization.

**Impact of lightweight strategies.** Tab. 5 conducts comparative analysis for two lightweight strategies, including Linear Attention (LA) and Pyramid Pooling Sampling (PPS). We further report the FLOPs and memory (using a pair of street image and satellite image as input). We can see that using no LA and PPS suffers from poor results, especially in cross-area performance. This is a reasonable result since vanilla dot-product attention is more prone to overfitting than linear attention, as well as much redundancy from multi-scale features interfering with information representation. But this problem is alleviated by a combination of two strategies, and leading to an interesting observation is that the performance gain of using two strategies is more pronounced in the two area evaluations while using fewer FLOPs and memory.

**Impact of hyperparameter.** Fig. 7 analyzes the impact of different dimension ratios of the global descriptor and local descriptor. As the proportion of dimensions of global descriptors relative to local descriptors increases, namely more global information, the same-area performance gradually improves while the cross-area performance notably enhances. Conversely, a decrease in same-area

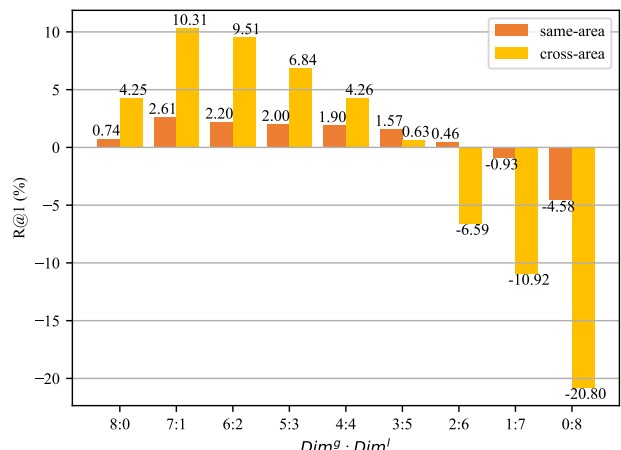

Figure 7: Impact of different descriptor dimensions. $Dim^g$ is dimension of Desc$^g$, and $Dim^l$ is dimension of Desc$^l$. We take the average R@1 score as a baseline, and all cases are subtracted from it, respectively.

results occurs slightly, but cross-area results rapidly decline. We also observe a severe deterioration when only local information is considered. This indicates that global information is crucial in image-to-image matching. However, relying on global information alone poses a limitation for performance. Therefore, it is indispensable to simultaneously learn both global and local information as our method for G2GAL'task. Based on the above results, we choose the ratio of dimensions between global and local descriptors to 7:1.

### 4.4 Visualization

**T-SNE visualization.** We use Sample4Geo and our MFRGN (both training on CVUSA) without **Pre-processing** and **Sampling** to obtain representation features of CVUSA and CVACT testing data. Fig. 8 visualizes the same-area and cross-area testing distribution by projecting the features onto a 2D space through t-SNE [37]. In Sample4Geo (a), the feature distributions of the same area and the cross area have a clear discrepancy. Our MFRGN (b) shows a coincidence between the cross-area distribution and the same-area distribution in both shape and scope. This proves that MFRGN can learn robust representation descriptors and effectively maps similar samples from same-area datasets and cross-area datasets to a similar feature representation space, which hence reduces the content distribution discrepancy between the two areas.

**Heatmap visualization.** Fig. 9 visualizes response maps generated by the two methods following the same settings as above. Sampel4Geo only focuses on certain local regions, owing to its pure CNN architecture. Our method allows the response to simultaneously concentrate on both global and local regions, such as the highway with a large coverage area and the local area of the roof. Such a property is attributed to the fact that MFRGN explicitly models both global and local features, allowing the model to achieve better representation ability against variations in cross-area scenarios.

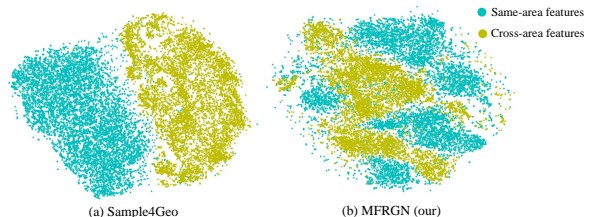

(a) Sample4Geo  (b) MFRGN (our)

Figure 8: T-SNE visualization of the same-area and cross-area testing distribution. The proposed MFRGN effectively maps similar samples from cross-area and same-area datasets to similar feature representation spaces and reduces the content distribution discrepancy between the two areas.

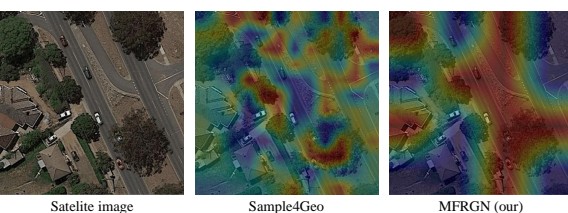

Satelite image  Sample4Geo  MFRGN (our)

Figure 9: Response visualization of Sample4Geo and the proposed MFRGN. MFRGN is more discriminative and its response is concentrated in both the global region and local region, e.g. the highway with a large coverage area and the local area of the roof.

## 5 CONCLUSION

In our work, we present a simple and effective method, i.e. MFRGN, to solve cross-area generalization for G2AGL by explicitly joint learning global and local representation from multi-scale features. We introduce a lightweight ViT-based Self and Cross Attention Module (SCAM) to efficiently learn global embedding from multi-scale features. Then, we utilize learned global embedding as prompts to boost local representation by the proposed Global-Prompt Attention Block (GPAB). Besides, we introduce a lightweight Transformer encoder and pyramid pooling sampling strategy to alleviate model burden and feature redundancy. Extensive experiments demonstrate competitive same-area performance and SOTA cross-area generalization of the proposed method without requiring strict assumptions that are unmet in real-life settings, or any task-specific techniques.

Future work will design a more effective multi-scale feature exploitation model to achieve better performance in terms of same-area and cross-area evaluation. We also aim to generalize this paradigm to other cross-view geo-localization frameworks, such as UAV-to-satellite geo-localization, which enjoys widespread popularity within the community of UAVs in Multimedia [1].

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
