# OpenReview forum: "MFRGN: Multi-scale Feature Representation Generalization Network For Ground-to-Aerial Geo-localization"
_acmmm.org/ACMMM/2024/Conference — MM2024 Poster_

### Official Review · Reviewer_UU8o · 2024-05-16

**Rating:** 3
**Confidence:** 3

**Summary:**

This paper aims to address the challenge of cross-area generalisation in ground-to-aerial geo-localization using a multi-scale feature representation model. Despite the effort to improve upon current methods, several issues undermine the effectiveness and clarity of the proposed solution.

**Strengths:**

The model is evaluated on three datasets and compared with some recent work. These comparisons show the model's performance for the place recognition task.

**Limitations:**

1- This paper primarily combines existing techniques without providing substantial innovation. The introduction of multi-scale features and the use of attention modules are well-explored areas, and the paper did not demonstrate how MFRGN significantly differs from or improves upon these established methods. Moreover, the paper does not provide adequate justification for the chosen architecture. For example, the reasons behind selecting a specific multi-scale (FPN) feature extraction method.

2- Some technical details are missing. For instance, the specifics of the multi-scale feature extraction process, the exact architecture of the attention modules, and the training procedures are inadequately described.

3- The authors should provide more comparisons, clearer justifications for their model design, and a better presentation of their methodology. Indeed, a clearer and more detailed presentation of the methodology, including algorithmic steps and parameter settings, would enhance the reproducibility of the work.

**Suitability:**

2

---

### Official Review · Reviewer_byhr · 2024-05-28

**Rating:** 3
**Confidence:** 2

**Summary:**

This manuscript introduces a multi-scale feature representation generalization framework for the ground-to-aerial geo-localization problem. Self and cross-attention modules and global-prompt attention blocks are proposed.

**Strengths:**

+ The performance of the proposed method, and its components seems to be decent, compared to its competitors.
+ The manuscript is well-organized.

**Limitations:**

+ I suggest the authors to re-organize the contribution part. "Our framework is simple..." can't count as a technical contribution.
+ Can Sample4Geo adapt with pre-cropping? If so, please report its performance under the same setting as MFRGN+.  The proposed MFRGN seems to have a similar pipeline to Sample4Geo, so it should be easy to do so.
+ Why does multiscale representation work so well for this task? I think this part of the manuscript is not well-motivated.
+ Please report parameters/GFLOPs comparison with previous works.

I'm willing to discuss in the rebuttal phase, and I will raise the score if my concerns are alleviated.

**Suitability:**

2

---

### Official Review · Reviewer_WWJy · 2024-05-28

**Rating:** 4
**Confidence:** 3

**Summary:**

This paper presents a multi-scale feature representation method for Ground-to-Aerial Geo-localization, especially for enhancing the cross-area generalization. The authors propose a global-local information module with two flows (global and local flows) to improve feature representation. Global flow yields global embeddings using a Self and Cross Attention module. Local flow uses a Global-Prompt Attention block to capture discriminative features with the global representation as prompts.

**Strengths:**

1. This paper is well-written and clearly presented.
2. The proposed method is novel and the contributions are sufficient. The proposed method achieves good performance in cross-area evaluation without task-specific pre-processing or hard sample mining.
3. The experimental setup is reasonable and the experimental results also seem good.

**Limitations:**

1. The proposed model architecture looks a bit complex compared to other methods, although the performance is good. Does this come at the cost of significantly increased inference runtime? It would be better to provide a comparison of inference runtime.
2. This proposed model seems to be directly applicable to UAV-to-satellite geo-localization, but it is considered as future work in this paper. I'm confused about this. How does the proposed method perform on the University-1652 dataset?
3. Although methods that use both global and local features are rare in cross-view geo-localization, they are indeed common in visual geo-localization (a.k.a. place recognition) [1][2][3]. It should be mentioned in Related Work.

[1] Transvpr: Transformer-based place recognition with multi-level attention aggregation. CVPR 2022.

[2] Towards Seamless Adaptation of Pre-trained Models for Visual Place Recognition. ICLR 2024.

[3] Deep Homography Estimation for Visual Place Recognition. AAAI 2024.

**Suitability:**

3

---

### Meta-Review · Area_Chair_A5Vt · 2024-06-29

**Recommendation:** Accept (Poster)
**Confidence:** 4

**Metareview:**

Well written paper with a reasonable technical quality and convincing results.